# Association between tocilizumab, sarilumab and all-cause mortality at 28 days in hospitalised patients with COVID-19: A network meta-analysis

Peter J. Godolphin[1]*, David J. Fisher[1], Lindsay R. Berry[2], Lennie P. G. Derde[3,4], Janet V. Diaz[5], Anthony C. Gordon[6], Elizabeth Lorenzi[2], John C. Marshall[7], Srinivas Murthy[8], Manu Shankar-Hari[9,10], Jonathan A. C. Sterne[11,12,13], Jayne F. Tierney[1], Claire L. Vale[1]

1 MRC Clinical Trials Unit at University College London, Institute of Clinical Trials and Methodology, London, United Kingdom, 2 Berry Consultants, Austin, Texas, United States of America, 3 Department of Intensive Care Medicine, University Medical Center Utrecht, Utrecht, The Netherlands, 4 University Medical Center Utrecht, Julius Center for Health Sciences and Primary Care, Utrecht, The Netherlands, 5 Clinical Unit, Health Emergencies Programme, World Health Organization, Geneva, Switzerland, 6 Division of Anaesthetics, Pain Medicine and Intensive Care, Imperial College London, London, United Kingdom, 7 Li Ka Shing Knowledge Institute, St Michael's Hospital, University of Toronto, Toronto, Ontario, Canada, 8 Department of Pediatrics, University of British Columbia, Vancouver, Canada, 9 Centre for Inflammation Research, The University of Edinburgh, Edinburgh, United Kingdom, 10 Department of Critical Care Medicine, Royal Infirmary of Edinburgh, NHS Lothian, Edinburgh, United Kingdom, 11 Population Health Sciences, Bristol Medical School, University of Bristol, Bristol, United Kingdom, 12 NIHR Bristol Biomedical Research Centre, Bristol, United Kingdom, 13 Health Data Research UK South West, Bristol, United Kingdom

* p.godolphin@ucl.ac.uk

**Data Availability Statement:** All relevant data are within the article and its Supporting information files.

## Abstract

### Background

A recent prospective meta-analysis demonstrated that interleukin-6 antagonists are associated with lower all-cause mortality in hospitalised patients with COVID-19, compared with usual care or placebo. However, emerging evidence suggests that clinicians are favouring the use of tocilizumab over sarilumab. A new randomised comparison of these agents from the REMAP-CAP trial shows similar effects on in-hospital mortality. Therefore, we initiated a network meta-analysis, to estimate pairwise associations between tocilizumab, sarilumab and usual care or placebo with 28-day mortality, in COVID-19 patients receiving concomitant corticosteroids and ventilation, based on all available direct and indirect evidence.

### Methods

Eligible trials randomised hospitalised patients with COVID-19 that compared tocilizumab or sarilumab with usual care or placebo in the prospective meta-analysis or that directly compared tocilizumab with sarilumab. Data were restricted to patients receiving corticosteroids and either non-invasive or invasive ventilation at randomisation.

Pairwise associations between tocilizumab, sarilumab and usual care or placebo for all-cause mortality 28 days after randomisation were estimated using a frequentist contrast-

**Funding:** This study was funded by Prostate Cancer UK in the form of funds to PJG and DJF [RIA 16-ST2-020], by the National Institute for Health Research in the form of funds to PJG [NIHR301653], by the Medical Research Council in the form of funds to DJF, JFT, and CLV [MC_UU_00004/06], by the National Institute for Health Research in the form of funds to MS-H [NIHR-CS-2016-16-011], and by the National Institute for Health Research in the form of funds to ACG [RP-2015-06-018].

**Competing interests:** LPGD is a member of the COVID-19 guideline committee for the Society of Critical Care Medicine/European Society of Intensive Care Medicine/Surviving Sepsis Campaign. ACG has received personal fees from Thirty Respiratory Ltd and GlaxoSmithKline. JCM received personal fees from AM Pharma (for serving as the chair of a data and safety monitoring board), Gilead (for serving as a consultant), and Critical Care Medicine (for serving as associate editor). This does not alter our adherence to PLOS ONE policies on sharing data and materials.

based network meta-analysis of odds ratios (ORs), implementing multivariate fixed-effects models that assume consistency between the direct and indirect evidence.

## Findings

One trial (REMAP-CAP) was identified that directly compared tocilizumab with sarilumab and supplied results on all-cause mortality at 28-days. This network meta-analysis was based on 898 eligible patients (278 deaths) from REMAP-CAP and 3710 eligible patients from 18 trials (1278 deaths) from the prospective meta-analysis. Summary ORs were similar for tocilizumab [0·82 [0·71–0·95, p = 0·008]] and sarilumab [0·80 [0·61–1·04, p = 0·09]] compared with usual care or placebo. The summary OR for 28-day mortality comparing tocilizumab with sarilumab was 1·03 [95%CI 0·81–1·32, p = 0·80]. The p-value for the global test of inconsistency was 0·28.

## Conclusions

Administration of either tocilizumab or sarilumab was associated with lower 28-day all-cause mortality compared with usual care or placebo. The association is not dependent on the choice of interleukin-6 receptor antagonist.

## Introduction

Following the recent publication of results from a prospective meta-analysis [1] and an updated guideline from the WHO [2], the interleukin-6 receptor antagonists, tocilizumab and sarilumab, have been recommended alongside corticosteroids for the routine treatment of hospitalised patients requiring oxygen support for COVID-19.

Findings from the prospective meta-analysis, which unlike standard meta-analysis is planned whilst trials are ongoing, preceding any knowledge of trial results and therefore less prone to biases sometimes associated with standard meta-analysis of aggregate data [3], showed that the interleukin-6 antagonists were associated with lower all-cause mortality 28 days after randomisation than standard care alone. In a prespecified analysis stratified by individual interleukin-6 receptor antagonists, whilst there was a clear association between reduced mortality and tocilizumab (based on the results of 8048 patients from 19 randomised trials), the evidence supporting the use of sarilumab (based on 2826 patients from 9 randomised trials) was less certain. In further pre-specified analyses, a stronger association between the interleukin-6 antagonists and reduced mortality was observed among patients receiving concomitant corticosteroids at randomisation than those not receiving corticosteroids, and the proportion of patients receiving concomitant corticosteroids at randomisation was lower in sarilumab trials than tocilizumab trials. If, based on these findings, clinicians and healthcare providers tend to favour the use of tocilizumab, there will inevitably be implications on demand and availability, potentially limiting patient access to tocilizumab.

The best way to resolve this uncertainty is to compare the relative effectiveness of tocilizumab with sarilumab. However, because the prospective meta-analysis set out to compare interleukin-6 antagonists with standard of care, trials that directly compared individual agents were excluded. Therefore, only indirect comparisons between tocilizumab and sarilumab, summarised as a ratio of odds ratios, were possible. An indirect comparison of the two agents, in patients receiving corticosteroids as part of usual care, suggested similar associations for both

agents with 28-day all-cause mortality (Ratio of odds ratios, 0·77 [95%CI 0·44–1·33, p = 0·34]), but this comparison was not precisely estimated. To better compare the effectiveness of these two agents, direct randomised comparisons are needed. The Randomised, Embedded, Multi-factorial Adaptive Platform Trial for Community-Acquired Pneumonia (REMAP-CAP) trial, which randomised critically ill patients with COVID-19 requiring either non-invasive ventilation (NIV) or invasive mechanical ventilation (IMV) including Extracorporeal Membrane Oxygenation (ECMO) [4] is, to date, the only reported randomised clinical trial that has directly compared tocilizumab and sarilumab [5]. Analysed as part of the immune modulation therapy domain of the trial, pre-defined triggers for equivalence between tocilizumab and sarilumab were met. The investigators reported beneficial effects of both tocilizumab and sarilumab on the primary outcome, organ support-free days, as well as on all pre-planned secondary outcomes including in hospital survival; 90-day survival; and both intensive care unit and hospital discharge. Furthermore, they reported that in their Bayesian analysis, the probability that sarilumab was non-inferior to tocilizumab was 98.9%.

Therefore, to inform clinical practice more fully and to clarify the evidence regarding these two treatments, we planned a network meta-analysis, bringing together the relevant data on tocilizumab and sarilumab from all randomised clinical trials. The aim of this new analysis is to estimate the pairwise associations between administration of tocilizumab, sarilumab or usual care or placebo and 28-day mortality, in COVID-19 patients receiving concomitant corticosteroids and NIV, IMV or ECMO, based on all the available direct and indirect evidence.

## Methods

This network meta-analysis is reporting according to the Preferred Reporting Items for Systematic Reviews and Meta-Analyses extension statement to Network Meta-Analyses [6] (see S1 Checklist).

Eligible randomised trials that aimed to compare tocilizumab or sarilumab with standard care in the treatment of hospitalised patients with COVID-19 were identified from the searches conducted by the same authors for a recently published systematic review and prospective meta-analysis [1]. Full details of the methods used have been previously reported [1], and are included in the prospectively registered protocol (CRD42021230155) [7].

For this network meta-analysis, we also carried out searches of trial registers (Clinicaltrials. gov and the EU Clinical Trials Register, most recent search 27[th] August 2021) to identify any randomised trials in addition to REMAP-CAP that directly compared tocilizumab with sarilumab in a similar population, using the search terms sarilumab, tocilizumab, random* and COVID. Patients were eligible for inclusion in this network meta-analysis if they were included in any of the eligible randomised trials and received either NIV (including high-flow nasal canula), IMV or ECMO at randomisation. Furthermore, because the prospective meta-analysis demonstrated that corticosteroid use modifies the association of interleukin-6 antagonists with mortality, patients also needed to have received corticosteroids as part of usual care to be eligible.

The primary outcome was all-cause mortality up to 28 days after randomisation. Data on all eligible patients included in the prospective meta-analysis were extracted from the summary data supplied. We requested data using bespoke data collection forms (developed for the prospective meta-analysis) for any trials identified as having made a direct comparison between tocilizumab and sarilumab.

All included trials secured institutional review board approval, and informed consent for participation in each trial was obtained, consistent with local institutional review board

requirements. Approval was not required for these secondary analyses as all data were published either as part of the prospective meta-analysis and/or in individual trial reports.

## Risk of bias

Risk of bias for each trial included in the prospective meta-analysis had already been assessed for all-cause mortality 28 days after randomisation as part of the prospective meta-analysis and was not repeated here. We planned to similarly assess risk of bias for any additional eligible trials identified for the network meta-analysis for this outcome using version 2 of the Cochrane Risk of Bias Assessment Tool [8].

## Contemporaneous randomisation in REMAP-CAP

Because REMAP-CAP is a multi-arm trial with an adaptive non-parallel design, for the purposes of this analysis it is represented as three independent observations in the model (tocilizumab vs usual care or placebo, sarilumab vs usual care or placebo, sarilumab vs tocilizumab). A small group of patients (21, 4 deaths by 28 days) were randomised to usual care or placebo contemporaneously with both treatment arms. We re-allocated these patients (and events) to the tocilizumab vs usual care or placebo and sarilumab versus usual care or placebo observations in proportion to their total counts and events, and thereafter assumed independence between these observations.

## Statistical analysis

Pairwise associations between tocilizumab, sarilumab and usual care or placebo were estimated using a network meta-analysis of odds ratios (ORs), using a frequentist contrast-based approach implemented in multivariate fixed-effects meta-analysis models [9]. These models assume consistency between 'direct evidence' (associations estimated in trials directly comparing the pair of interventions) and 'indirect evidence' (associations estimated through the network). The 'net evidence' from the network meta-analysis is a weighted average of the direct and indirect evidence. Inconsistency between direct and indirect evidence was examined locally using symmetrical node-splitting [10] and globally using a design-by-treatment interaction model [9,11]. Borrowing of strength statistics were calculated using the score decomposition method [12] to illustrate the proportion of information for each net estimate that is due to indirect evidence. Treatment rankings were also calculated and are summarised according to the surface under the cumulative ranking curve (SUCRA) value, which represents the re-scaled mean ranking [13,14]. Following the approach in the prospective meta-analysis [1], we report precise p values and do not set a threshold for statistical significance. The certainty of evidence in each comparison was rated following the GRADE approach to network meta-analysis [15,16], with this completed independently by two reviewers [PJG and CLV] and any discrepancies resolved through discussion. The certainty of evidence was rated as either High, Moderate, Low or Very Low. All analyses were conducted in Stata statistical software version 16.1 [StataCorp, USA] using the 'network' user-written command suite [17].

## Results

### Study selection and description of eligible trials

Of the 27 trials included in the prospective meta-analysis, nine randomised patients prior to guidance to include corticosteroids as part of routine care, or excluded patients requiring non-invasive or mechanical ventilation, or used an interleukin-6 agent other than tocilizumab or sarilumab. Thus, these trials are ineligible for the network meta-analysis (Fig 1). Of the nine

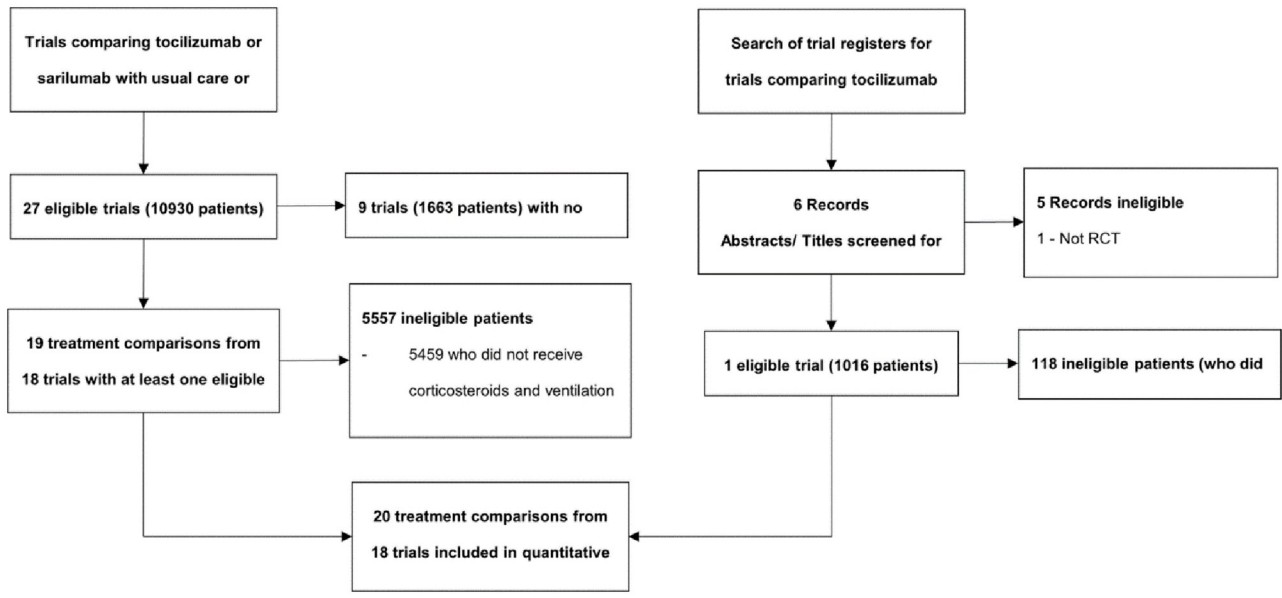

**Fig 1. Flow diagram showing the identification of eligible trials and patients.**

trials, a similar number compared tocilizumab (5 trials, 848 patients) to usual care or placebo as compared sarilumab (4 trials, 815 patients) to usual care or placebo. The remaining 18 trials contained at least one eligible patient and are included in this network meta-analysis. 14 trials are published [4,18–28], one is reported on a pre-print server [29] and three are not yet published (see Table 1 for trial registration numbers). These 18 trials had compared tocilizumab (13 trials) or sarilumab (4 trials) or both (1 trial) with usual care or placebo and include 3710 patients (40%; 1278 deaths by 28 days) who received corticosteroids and either non-invasive or mechanical ventilation and were therefore eligible for inclusion in the network meta-analysis (Fig 1).

Searches of trial registers for eligible randomised trials that had directly compared tocilizumab with sarilumab in a similar patient population did not return any further trials in addition to the recently published REMAP-CAP trial [5]. Full details of search results are given in Fig 1. Trial investigators for REMAP-CAP obtained approval from the International Trial Steering Committee to supply data for this analysis. Data were supplied on June 12th, 2021 using a standardised outcome data collection form (developed for use in the prospective meta-analysis) and finalised data were subsequently verified by the trial team prior to inclusion in this analysis. Of 1018 patients from the REMAP-CAP trial who were randomised to receive either tocilizumab or sarilumab, 898 (88%; 278 deaths by 28 days) received NIV, IMV or ECMO plus corticosteroids at randomisation and were eligible for inclusion in the network meta-analysis.

## Risk of bias within studies

Detailed risk of bias assessments for the 18 included trials that contributed to the prospective meta-analysis have already been reported [1]. In summary, 12 were assessed as low risk of bias (1003 deaths by 28 days, 65% of total deaths); five were judged to have some concerns (257 deaths by 28 days, 17% of total deaths) largely as small numbers of patients who did not receive their assigned interventions were excluded. One trial (18 deaths by 28 days, 1% of total deaths) was judged as high risk of bias as the usual procedures to ensure concealment of the allocation sequence were not in place; however, concealed allocation did appear to have been

**Table 1. Summary of included trials, patient characteristics and all-cause mortality 28 days after randomisation.**

| Trial name | Trial registration No. | No. of eligible patients / total randomised | For eligible patients, concomitant therapy at randomisation (%) | | | For eligible patients, 28-day mortality (Deaths / Patients) | | |
|---|---|---|---|---|---|---|---|---|
| | | | Corticosteroids | Non-invasive ventilation | Invasive mechanical ventilation | Usual care or Placebo | Tocilizumab | Sarilumab |
| **Tocilizumab versus usual care or placebo** | | | | | | | | |
| ARCHITECTS | NCT04412772 | 19/21 | 19 (100%) | 0 (-) | 19 (100%) | 1/10 | 0/9 | |
| CORIMUNO-TOCI-ICU | NCT04331808 | 12/92 | 12 (100%) | 3 (25%) | 9 (75%) | 2/4 | 4/8 | |
| COV-AID | NCT04330638 | 42/153 | 42 (100%) | 31 (74%) | 11 (26%) | 3/20 | 5/22 | |
| COVACTA | NCT04320615 | 69/438 | 69 (100%) | 27 (39%) | 42 (61%) | 11/26 | 13/43 | |
| COVIDOSE2-SS-A | NCT04479358 | 1/27 | 1 (100%) | 1 (100%) | 0 (-) | 0/1 | 0/0 | |
| COVIDSTORM | NCT04577534 | 10/39 | 10 (100%) | 10 (100%) | 0 (-) | 0/4 | 0/6 | |
| EMPACTA | NCT04372186 | 94/377 | 94 (100%) | 94 (100%) | 0 (-) | 7/33 | 13/61 | |
| HMO-020-0224 | NCT04377750 | 46/54 | 46 (100%) | 19 (41%) | 27 (59%) | 8/15 | 10/31 | |
| ImmCoVA | EudraCT 2020-001748-24 | 29/49 | 29 (100%) | 29 (100%) | 0 (-) | 2/18 | 2/11 | |
| PreToVid | EudraCT 2020-001375-32 | 82/354 | 82 (100%) | 79 (96%) | 3 (4%) | 12/43 | 8/39 | |
| RECOVERY | NCT04381936 | 1849/4116 | 1849 (100%) | 1444 (78%) | 405 (22%) | 427/954 | 356/895 | |
| REMAP-CAP (a) | NCT02735707 | 429/711 | 429 (100%) | 314 (73%) | 115 (27%) | 70/201[a] | 53/213[a] | |
| REMDACTA | NCT04409262 | 523/640 | 523 (100%) | 445 (85%) | 78 (15%) | 39/179 | 68/344 | |
| TOCIBRAS | NCT04403685 | 31/129 | 31 (100%) | 20 (65%) | 11 (35%) | 5/20 | 7/11 | |
| **Sarilumab versus usual care or placebo** | | | | | | | | |
| CORIMUNO-SARI-ICU | NCT04324073 | 2/81 | 2 (100%) | 1 (50%) | 1 (50%) | 0/2 | | 0/0 |
| REGENERON-P2 | NCT04315298 | 63/457 | 63 (100%) | 19 (30%) | 44 (70%) | 4/10 | | 26/53 |
| REGENERON-P3 | NCT04315298 | 328/1330 | 328 (100%) | 178 (54%) | 150 (46%) | 21/71 | | 79/257 |
| REMAP-CAP (b) | NCT02735707 | 96/113 | 96 (100%) | 84 (88%) | 12 (13%) | 13/49[a] | | 8/44[a] |
| SARCOVID | NCT04357808 | 3/30 | 3 (100%) | 3 (100%) | 0 (-) | 0/0 | | 1/3 |
| **Tocilizumab versus sarilumab** | | | | | | | | |
| REMAP-CAP (c) | NCT02735707 | 898/1018 | 898 (100%) | 596 (66%) | 302 (34%) | | 169/529 | 109/369 |

[a]REMAP-CAP has a small group of patients and events (21, 4 deaths by 28 days) randomised to usual care or placebo contemporaneously with both Tocilizumab and Sarilumab. These patients are events are re-allocated in proportion to their total counts and events.

implemented as intended. Risk of bias for the additional REMAP-CAP direct comparison was judged as low risk of bias. Thus, in total, 12 trials (14 comparisons, 1281 deaths by 28 days, 82% of total deaths) were judged as low risk of bias.

## Synthesis of results

The direct comparison with the greatest amount of information was tocilizumab versus usual care or placebo (Fig 2, 3221 patients, 1126 deaths by 28 days). There was relatively little information for the direct comparison of sarilumab with usual care or placebo (489 patients, 152 deaths by 28 days). The direct comparison of tocilizumab with sarilumab was from REMAP-CAP (898 patients, 278 deaths by 28 days). Fig 3 presents the direct evidence for each of the included trials. For both the tocilizumab versus usual care or placebo and sarilumab versus usual care or placebo comparisons, a single trial contributed approximately two-thirds of the information for the direct estimate (RECOVERY for the tocilizumab comparison and REGENERON-P3 for the sarilumab comparison).

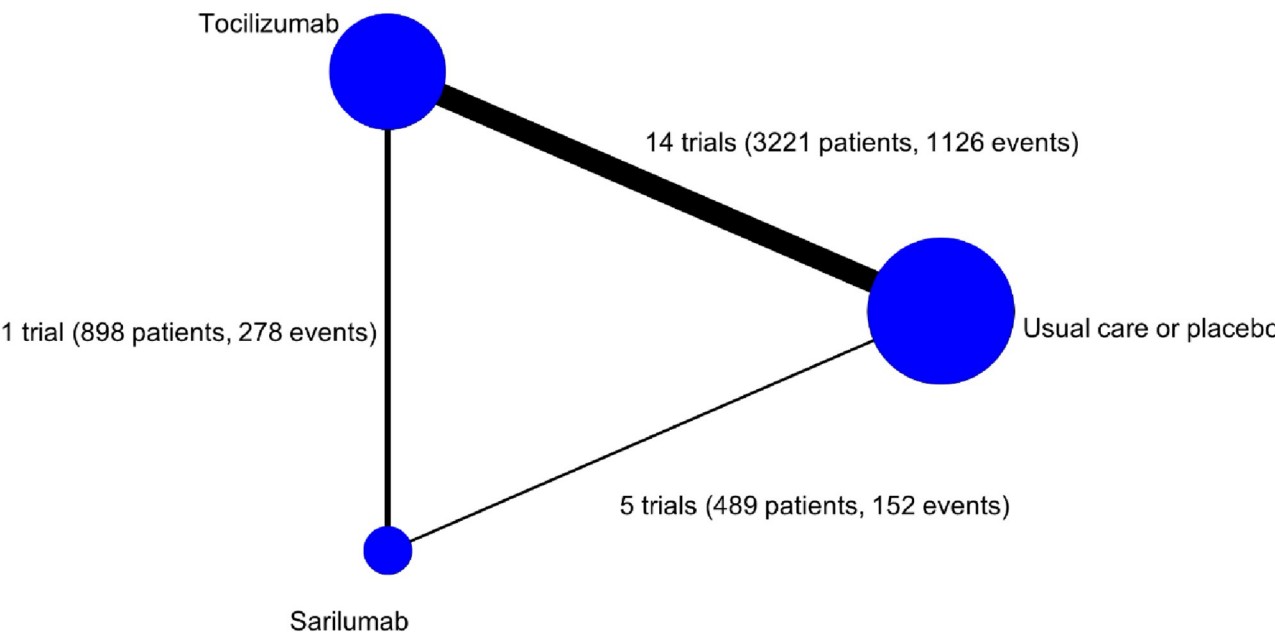

**Fig 2. Network map showing numbers of trials in each direct treatment comparison.** The node size is proportional to the number of trials that include this treatment. The width of the lines is proportional to the total number of events involved in each direct comparison.

Based on the network meta-analysis, the net ORs for 28-day mortality were similar for tocilizumab [95%CI 0·82 [0·71–0·95, p = 0·008]] and sarilumab [95%CI 0·80 [0·61–1·04, p = 0·09]] compared with usual care or placebo (Table 2, Fig 4), although the tocilizumab comparison borrowed less strength from the network (borrowing of strength 7%) than the sarilumab comparison (borrowing of strength 67%). The net OR for 28-day mortality comparing tocilizumab with sarilumab was 1·03 [95%CI 0·81–1·32, p = 0·80], with this comparison borrowing 26% of strength from the network. The global p value for inconsistency was 0·28. Both tocilizumab and sarilumab were ranked similarly with high SUCRA values (70% and 78% respectively, Table 3). Usual care or placebo had a 95% probability of being the least effective treatment. The certainty of evidence for each comparison is displayed in S1 Table. Tocilizumab versus usual care was rated as High, with sarilumab versus usual care and tocilizumab versus sarilumab both rated as Moderate (both downgraded due to Imprecision).

## Discussion

In this network meta-analysis of patients receiving both corticosteroids and either NIV, IMV or ECMO at randomisation, both tocilizumab and sarilumab were associated with lower all-cause mortality 28 days after randomisation compared with usual care or placebo. The associations of these agents with all-cause mortality appeared similar, consistent with the direct findings from the REMAP-CAP trial [5] in which tocilizumab and sarilumab met the criteria for equivalence. More generally, these results confirm a clear association of interleukin-6 receptor antagonists with lower all-cause mortality in this patient population.

The comparison of tocilizumab versus usual care or placebo was based mainly on direct evidence from the prospective meta-analysis, with only limited additional information from the network. By contrast, for sarilumab versus usual care or placebo, the direct comparison was limited to fewer than 500 patients from the prospective meta-analysis. Therefore, the indirect evidence (arising from the association of tocilizumab with reduced all-cause mortality

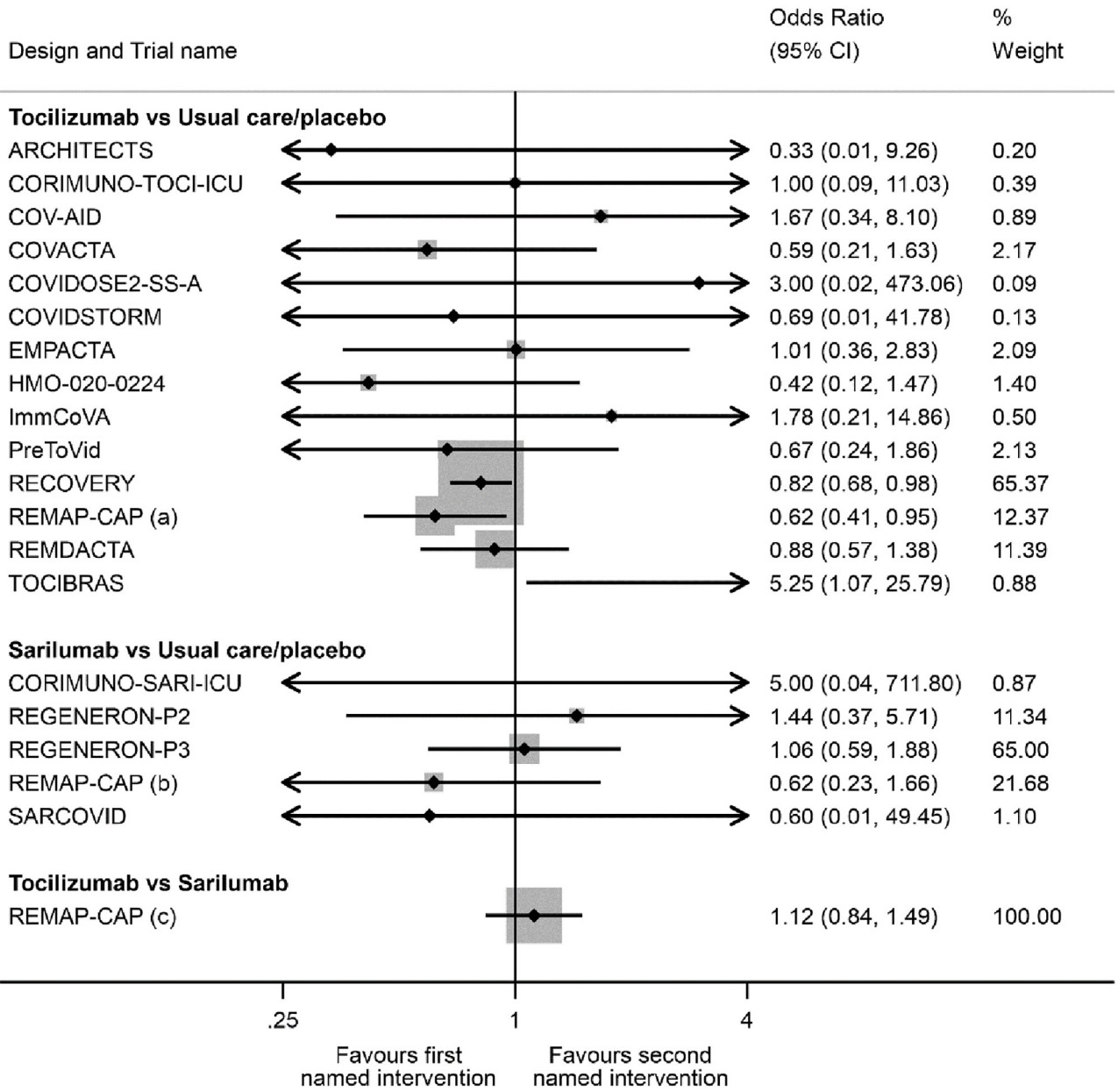

**Fig 3. Summary of the direct evidence from each included trial for all-cause mortality 28 days after randomisation.** The % weight corresponds to the contribution each trial makes to the pooled direct evidence for each treatment comparison.

compared to usual care or placebo and the direct comparison of tocilizumab with sarilumab) has a substantial impact on the net estimate for this comparison. In the absence of any other direct comparisons of tocilizumab with sarilumab, this network meta-analysis provides the strongest evidence in support of the hypothesis that both agents are similarly associated with lower all-cause mortality at 28-days in this patient population.

A separate living network meta-analysis found that both tocilizumab and sarilumab added to usual care including corticosteroids "probably reduce mortality" in patients with severe or critical COVID-19 [30] based on results for all patients from 36 randomised trials, irrespective of corticosteroid use or extent of oxygen support at randomisation. Including all patients

**Table 2. Summary of direct, indirect, and net evidence for the associations of tocilizumab, sarilumab and usual care or placebo with all-cause mortality 28 days after randomisation for patients receiving corticosteroids and either NIV, IMV or ECMO at randomisation.**

| Comparison | Number of trials | Deaths / patients from direct evidence | | OR (95% CI), p from direct evidence | OR (95% CI), p from indirect evidence | Net OR (95% CI), p from network meta-analysis | Inconsistency p value |
|---|---|---|---|---|---|---|---|
| | | Intervention 1[a] | Intervention 2[a] | | | | |
| Tocilizumab vs usual care or placebo | 14 | 539/1693 | 587/1528 | 0·80 (0·69, 0·93), p = 0·004 | 1·10 (0·64, 1·90), p = 0·74 | 0·82 (0·71, 0·95), p = 0·008 | p = 0·28 |
| Sarilumab vs usual care or placebo | 5 | 114/357 | 38/132 | 0·98 (0·62, 1·56), p = 0·94 | 0·72 (0·52, 0·99), p = 0·05 | 0·80 (0·61, 1·04), p = 0·09 | |
| Tocilizumab vs sarilumab | 1 | 169/529 | 109/369 | 1·12 (0·84, 1·49), p = 0·44 | 0·82 (0·50, 1·33), p = 0·42 | 1·03 (0·81, 1·32), p = 0·80 | |

NIV: Non-invasive ventilation. IMV: Invasive mechanical ventilation. ECMO: Extracorporeal Membrane Oxygenation.

Note, the REMAP-CAP trial contributes to all three comparisons for each network.

[a]Intervention 1 refers to the treatment listed first, while Intervention 2 is the treatment listed second. For example, for the comparison of tocilizumab versus usual care or placebo, Intervention 1 is tocilizumab and Intervention 2 is usual care or placebo.

Local tests for inconsistency, p = 0·28 for all three comparisons.

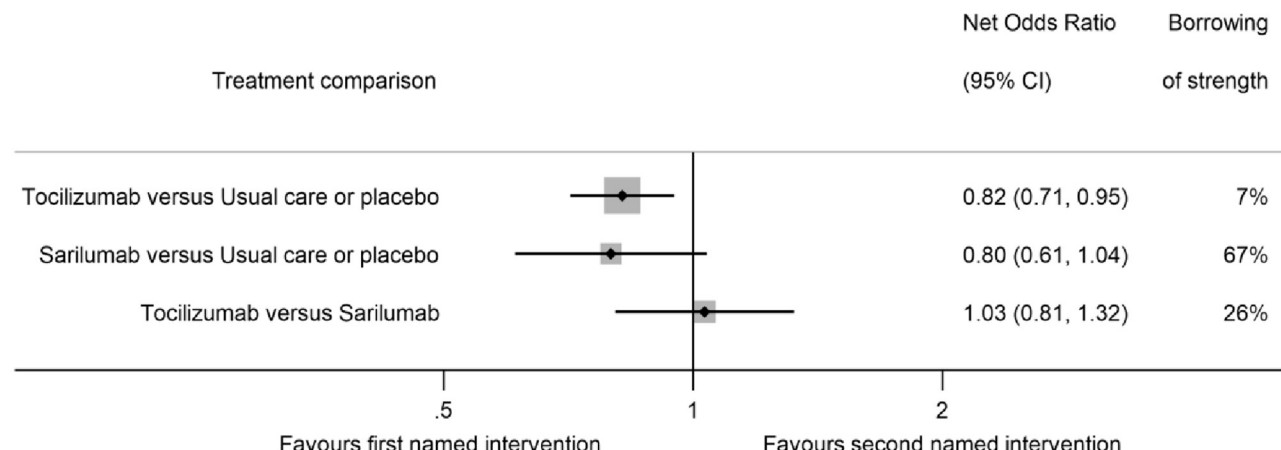

**Fig 4. Network associations of tocilizumab, sarilumab and usual care or placebo for patients receiving corticosteroids and either NIV, IMV or ECMO at randomisation with all-cause mortality 28 days after randomisation.** NIV: Non-invasive ventilation. IMV: Invasive mechanical ventilation. ECMO: Extracorporeal Membrane Oxygenation. Size of markers is proportional to the inverse of the variance from the net estimate. Borrowing of strength illustrates the proportion of information for each net odds ratio that is due to indirect evidence.

**Table 3. Ranking of interventions (% probability) and SUCRA values for all-cause mortality 28 days after randomisation.**

| Rank | Sarilumab | Tocilizumab | Usual care or placebo |
|---|---|---|---|
| Best | 59.7 | 40.1 | 0.1 |
| Second | 35.5 | 59.5 | 5.0 |
| Worst | 4.8 | 0.3 | 94.9 |
| **SUCRA** | 78% | 70% | 26% |

SUCRA: Surface under the cumulative ranking curve.

increases the possibility of inconsistency between patients included in the indirect and direct comparisons of tocilizumab and sarilumab, as patients in the direct comparison (i.e. from the REMAP-CAP trial), all received both corticosteroids and invasive or non-invasive ventilation. The authors also down-rated the certainty of evidence for tocilizumab versus usual care, sarilumab versus usual care and tocilizumab versus sarilumab, as Moderate, Low and Low respectively, based on a lack of blinding in many of the included trials.

In contrast, we restricted the network to the subset of patients receiving both oxygen support and corticosteroids, making them more comparable with each other and to the REMAP-CAP direct tocilizumab and sarilumab comparison. Therefore, variability of the population within the network and resulting inconsistency was reduced, and interpretability was increased. This was only possible through the prospective and collaborative approach [3] we adopted as part of the prospective meta-analysis [1], collecting detailed data on both oxygen support and corticosteroid use subgroups. This enabled us to make decisions often only available in an individual participant data meta-analysis [31] and resulted in increased consistency and harmonisation. Furthermore, given that over 80% of the total events included in this network were from trials judged to be at low risk of bias, and because of a lack of subjectivity in the assessment of a mortality outcome, we have kept our risk of bias assessments consistent with the approach used in the prospective meta-analysis [1] and by the WHO guideline panel [2]. Consequently, we rated the certainty of evidence as High, Moderate and Moderate for the comparisons of tocilizumab versus usual care, sarilumab versus usual care and tocilizumab versus sarilumab, respectively.

This study has some limitations. First, because these results are focused on patients treated with corticosteroids and either NIV, IMV or ECMO, alongside interleukin-6 receptor antagonists, they may not be generalised to less critically ill patients or to those not receiving steroids or non-invasive or mechanical ventilation. Second, the direct evidence in each of the three comparisons included in this network meta-analysis came predominantly from a single trial (either RECOVERY, REGENERON-P3 or REMAP-CAP), with these three trials primarily conducted in high income countries. With the direct evidence limited to only high-income counties we cannot be certain how their results might translate into lower income settings. Third, four of the included trials have not yet been published in peer-reviewed journals, either available as pre-print publications or currently unpublished. However, thorough checking and verification was carried out as part of the original prospective meta-analysis procedure, with the same process applied to the new REMAP-CAP trial data, and we have no concerns about the conduct or quality of the data from any of the as yet unpublished trials.

In conclusion, this network meta-analysis of clinical trials of hospitalised patients with COVID-19 receiving ventilation and corticosteroids at randomisation, confirms that administration of tocilizumab or sarilumab, compared with usual care or placebo, is associated with similarly lower 28-day all-cause mortality.

## Supporting information

**S1 Checklist. PRISMA NMA checklist of items to include when reporting a systematic review involving a network meta-analysis.**
(DOCX)

**S1 Table. Certainty assessment for each comparison.**
(DOCX)

## Author Contributions

**Conceptualization:** Peter J. Godolphin, David J. Fisher, Manu Shankar-Hari, Jonathan A. C. Sterne, Claire L. Vale.

**Data curation:** Peter J. Godolphin, David J. Fisher, Lindsay R. Berry, Lennie P. G. Derde, Anthony C. Gordon, Elizabeth Lorenzi, Manu Shankar-Hari, Jonathan A. C. Sterne, Claire L. Vale.

**Formal analysis:** Peter J. Godolphin, David J. Fisher, Jonathan A. C. Sterne.

**Investigation:** Peter J. Godolphin, David J. Fisher, Lindsay R. Berry, Lennie P. G. Derde, Janet V. Diaz, Anthony C. Gordon, Elizabeth Lorenzi, John C. Marshall, Srinivas Murthy, Manu Shankar-Hari, Jonathan A. C. Sterne, Jayne F. Tierney, Claire L. Vale.

**Supervision:** Janet V. Diaz, John C. Marshall, Srinivas Murthy, Manu Shankar-Hari, Jonathan A. C. Sterne, Jayne F. Tierney, Claire L. Vale.

**Writing – original draft:** Peter J. Godolphin, David J. Fisher, Jonathan A. C. Sterne, Claire L. Vale.

**Writing – review & editing:** Peter J. Godolphin, David J. Fisher, Lindsay R. Berry, Lennie P. G. Derde, Janet V. Diaz, Anthony C. Gordon, Elizabeth Lorenzi, John C. Marshall, Srinivas Murthy, Manu Shankar-Hari, Jonathan A. C. Sterne, Jayne F. Tierney, Claire L. Vale.

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
