## [Decision Letter · Decision Letter 0]

21 Apr 2022

PONE-D-22-07351Association between tocilizumab, sarilumab and all-cause mortality at 28 days in hospitalised patients with COVID-19: A network meta-analysisPLOS ONE

Dear Dr. Godolphin,

Thank you for submitting your manuscript to PLOS ONE. After careful consideration, we feel that it has merit but does not fully meet PLOS ONE’s publication criteria as it currently stands. Therefore, we invite you to submit a revised version of the manuscript that addresses the points raised during the review process.

Both reviewers, experts in SR methodology, underlined the importance of the topic. However, both recommended the GRADE assessment of the certainty of evidence according to current standard for GRADE NMA. 

We look forward to receiving your revised manuscript.

Kind regards,

Andrea Cortegiani, M.D.

Academic Editor

PLOS ONE

Journal Requirements:

"LPGD is a member of the COVID-19 guideline committee for the Society of Critical Care Medicine/European Society of Intensive Care Medicine/Surviving Sepsis Campaign.

ACG has received personal fees from Thirty Respiratory Ltd and GlaxoSmithKline.

JCM received personal fees from AM Pharma (for serving as the chair of a data and safety monitoring board), Gilead (for serving as a consultant), and Critical Care Medicine (for serving as associate editor)."

Reviewers' comments:

Reviewer's Responses to Questions

**Comments to the Author**

1. Is the manuscript technically sound, and do the data support the conclusions?

Reviewer #1: Yes

Reviewer #2: Partly

2. Has the statistical analysis been performed appropriately and rigorously? 

Reviewer #1: Yes

Reviewer #2: Yes

3. Have the authors made all data underlying the findings in their manuscript fully available?

Reviewer #1: Yes

Reviewer #2: Yes

4. Is the manuscript presented in an intelligible fashion and written in standard English?

Reviewer #1: Yes

Reviewer #2: Yes

5. Review Comments to the Author

Reviewer #1: Thank you for the opportunity to review this rigorous and well-conducted network meta-analysis. I have only a few minor points for the authors to consider to enhance the interpretation of the study. One of the major rationales for addressing the question of tocilizumab vs. sarilumab is that clinicians tend to favour the former. The study partially addresses this question, but a few issues remain.

1) No evidence for this claim (page 3, lines 71-72) "Possibly related to interpretations of these findings, there is some emerging evidence to suggest that clinicians and healthcare providers have tended to favour the use of tocilizumab, with potential implications on patient access to these treatments." Anecdotally, this statements aligns with my experience but I am not sure whether it is true or not, and no citation is provided.

2) There was no consideration of the quality/certainty of evidence across studies, eg. GRADE. This is quite feasible to do even in the NMA setting (https://www.bmj.com/content/349/bmj.g5630). Consideration of imprecision, inconsistency, indirectness, incoherence, etc. in a structured way (I would advocate for GRADE) could provide the reader with a better sense of how confident we should be in these results. For instance the confidence intervals for the T vs. S comparisons are very wide in all 3 comparisons (direct, indirect, net); the potential for incoherence between direct vs. indirect T vs. S comparisons, etc. This is not merely a matter of academic interest, but also important in exploring the study question regarding why clinicians may favour T vs. S (eg. direct evidence is less precise, etc.)

3) Lastly, the discussion does lit tle to address the question which the NMA claims to address, re: clinician practice. While making such recommendations is more likely to be done in the setting of a guideline, it is important to note that the rationale for clinicians and patients choosing a treatment is based upon many factors (desirable effects; undesirable effects for sure; but also certainty of those effects, the balance, the feasibility, cost, acceptability etc.). While the review does not really address all these other issues, the discussion could note that these are also important, not just the mortality rate alone (as important as that is).

These are fairly minor concerns in a wonderful paper. Butt as it quite explicitly asks one question, I expected there to be greater exploration of the certainty of the effects of T vs. S and some discussion on the topic (including all the various factors for choosing a treatment which are not really addressed in this NMA).

Reviewer #2: The authors did not assess the certainty of the evidence, which- according to Cochrane- is considered a methodological standard for all systematic reviews. This can be done using GRADE or CINEMA They should not only add this assessment (which includes specifying the degree of contextualization they will use and their target of certainty), but incorporate it formally in the drawing of their conclusions regarding the effects of the interventions. They can find detailed information about certainty of evidence assessment process and drawing conclusions from NMA in the GRADE for NMA papers, making narrative statements/ conclusions in GRADE guidelines 26, and contextualization/ target of certainty rating in GRADE guidelines 32.

In addition, authors should follow the prisma statement for NMA for reporting.

6. PLOS authors have the option to publish the peer review history of their article (what does this mean?). If published, this will include your full peer review and any attached files.

Reviewer #1: No

Reviewer #2: **Yes: **Romina Brignardello-Petersen

---

## [Author Response · Author response to Decision Letter 0]

1 Jun 2022

Please see attached file for our response to reviewers

---

## [Decision Letter · Decision Letter 1]

15 Jun 2022

Association between tocilizumab, sarilumab and all-cause mortality at 28 days in hospitalised patients with COVID-19: A network meta-analysis

PONE-D-22-07351R1

Dear Dr. Godolphin,

We’re pleased to inform you that your manuscript has been judged scientifically suitable for publication and will be formally accepted for publication once it meets all outstanding technical requirements.

Kind regards,

Andrea Cortegiani, M.D.

Academic Editor

PLOS ONE

Additional Editor Comments (optional):

Reviewers' comments:

Reviewer's Responses to Questions

**Comments to the Author**

1. If the authors have adequately addressed your comments raised in a previous round of review and you feel that this manuscript is now acceptable for publication, you may indicate that here to bypass the “Comments to the Author” section, enter your conflict of interest statement in the “Confidential to Editor” section, and submit your "Accept" recommendation.

Reviewer #1: All comments have been addressed

2. Is the manuscript technically sound, and do the data support the conclusions?

Reviewer #1: Yes

3. Has the statistical analysis been performed appropriately and rigorously? 

Reviewer #1: Yes

4. Have the authors made all data underlying the findings in their manuscript fully available?

Reviewer #1: Yes

5. Is the manuscript presented in an intelligible fashion and written in standard English?

Reviewer #1: Yes

6. Review Comments to the Author

Reviewer #1: Thank you for the thoughtful and comprehensive responses to the reviewer comments. The manuscript has benefited from the use of a standardized reporting approach for the quality/certainty of evidence using GRADE. It is excellent and I have no further comments-- thank you.

7. PLOS authors have the option to publish the peer review history of their article (what does this mean?). If published, this will include your full peer review and any attached files.

Reviewer #1: **Yes: **Simon JW Oczkowski

---

## [Editor Report · Acceptance letter]

30 Jun 2022

PONE-D-22-07351R1 

Association between tocilizumab, sarilumab and all-cause mortality at 28 days in hospitalised patients with COVID-19: A network meta-analysis 

Dear Dr. Godolphin:

I'm pleased to inform you that your manuscript has been deemed suitable for publication in PLOS ONE. Congratulations! Your manuscript is now with our production department. 

Kind regards, 

on behalf of

Dr. Andrea Cortegiani 

Academic Editor

PLOS ONE